# Current Understanding of the HIF-1-Dependent Metabolism in Oral Squamous Cell Carcinoma

**DOI:** 10.3390/ijms21176083

**Published:** 2020-08-24

**Authors:** Alexander W. Eckert, Matthias Kappler, Ivo Große, Claudia Wickenhauser, Barbara Seliger

**Affiliations:** 1Klinik für Mund-, Kiefer- und Plastische Gesichtschirurgie, Universitätsklinik der Paracelsus Medizinischen Privatuniversität, Breslauer Str. 201, 90471 Nurnberg, Germany; 2Universitätsklinik und Poliklinik für Mund-, Kiefer- und Plastische Gesichtschirurgie, Martin-Luther-Universität Halle-Wittenebrg, Ernst- Grube-Straße 40, 06120 Halle, Germany; matthias.kappler@uk-halle.de; 3Institut für Informatik, Martin-Luther-Universität Halle-Wittenberg, Von-Seckendorff-Platz 1, 06120 Halle (Saale), Germany; ivo.grosse@informatik.uni-halle.de; 4Institut für Pathologie, Martin-Luther-Universität Halle-Wittenberg, Magdeburger Str. 14, 06112 Halle (Saale), Germany; claudia.wickenhauser@uk-halle.de; 5Institut für Medizinische Immunologie, Martin-Luther-Universität Halle-Wittenberg, Magdeburger Str. 14, 06112 Halle (Saale), Germany

**Keywords:** oral squamous cell carcinoma, metabolism, tumor microenvironment, hypoxia, prognosis

## Abstract

Oral squamous cell carcinoma (OSCC) is the 10th most frequent human malignancy and is thus a global burden. Despite some progress in diagnosis and therapy, patients’ overall survival rate, between 40 and 55%, has stagnated over the last four decades. Since the tumor node metastasis (TNM) system is not precise enough to predict the disease outcome, additive factors for diagnosis, prognosis, prediction and therapy resistance are urgently needed for OSCC. One promising candidate is the hypoxia inducible factor-1 (HIF-1), which functions as an early regulator of tumor aggressiveness and is a key promoter of energy adaptation. Other parameters comprise the composition of the tumor microenvironment, which determines the availability of nutrients and oxygen. In our opinion, these general processes are linked in the pathogenesis of OSCC. Based on this assumption, the review will summarize the major features of the HIF system-induced activities, its target proteins and related pathways of nutrient utilization and metabolism that are essential for the initiation, progression and therapeutic stratification of OSCC.

## 1. Introduction

Oral squamous cell carcinoma (OSCC) is the 10th most frequent human malignancy worldwide [1,2] with approximately 500,000 new cases each year [3]. During the last 20 years, the molecular, metabolic and immunologic features of this disease have been well characterized. These include some essential hallmarks in cancer development, such as sustained proliferation, evasion from growth suppressors, resistance to cell death, replicative immortality, enhanced angiogenesis, increased invasion and metastasis, as well as alterations in the cellular metabolism [4,5].

The pivotal role of hypoxia in different cancer types, including OSCC, suggests that the hypoxia-induced pathways are novel promising therapeutic targets [6]. In this context, it is noteworthy that a deregulated energy metabolism mediated by the hypoxia inducible factor 1α (HIF-1α), which functions as a master regulator to balance oxygen supply and demand, has been described in OSCC [5,7]. The consequences of an increased HIF-1α activity comprise an altered protein expression involved in angiogenesis, metabolic reprogramming, extracellular matrix (ECM) remodeling, epithelial mesenchymal transition (EMT), motility, invasion, metastasis, cancer stem cell maintenance, immune evasion, metabolic adaptations, as well as immunologic changes of the tumor microenvironment (TME) during early carcinogenesis, which have been recently reviewed in detail [3,8,9,10,11,12,13]. The activation of HIF-1α-regulated pathways caused the activation or inhibition of diverse proteins, as depicted in Figure 1. This minireview will address the possibilities of OSCC to adapt for enhanced growth, tumor progression and metastasis formation with the focus on the role of metabolic changes in these processes.

## 2. The HIF-1α System as a General Regulator of OSCC Progression

Since the induction of several signal transduction pathways is an early event in oral carcinogenesis prior to detectable clinical features, such as TNM and grading, potential biomarkers in tissue specimens as well as in whole saliva are urgently needed for the early detection of this disease [14]. In this context, it is noteworthy that the HIF pathway and HIF-dependent target proteins are involved in the progression and clinical outcome of OSCC [15,16,17,18,19].

Otto Warburg’s assumption of a general reprogramming process in tumors to switch their metabolism from the oxidative to the glycolytic status was published in London in 1930 [20,21]. Ninety years later, many insights into the complex HIF-1 system are available. The HIF-1 system is composed of the hypoxia-inducible HIF-1α and the constitutively expressed HIF-1β subunit [22]. At physiologic oxygen levels, the proline residues 402 and 564 of HIF-1α are hydroxylated by prolylhydroxylases, leading to an interaction with the von Hippel –Lindau (VHL) protein as an E3 ubiquitin ligase complex, which mediates rapid proteasomal degradation [23,24]. In addition, HIF-1α acetylation at Lys532 occurs under normoxia, which causes the degradation of HIF-1 [25,26]. In contrast, HIF-1α is stable under hypoxic conditions, transported to the nucleus and is conjugated there with the HIF-1β subunit, also called the Aryl Hydrocarbon Nuclear Translocator (ARNT) [27,28].

In combination with the transcriptional coactivator CBP/p300 complex and the hypoxia response elements (HREs) present in many gene promoters, HIF-1 is involved in the regulation of more than 1000 target genes [8] and is thus a central player in the transcriptional response to hypoxia. The target genes controlled by HIF-1α are related to angiogenesis, cell proliferation/survival as well as glucose and iron metabolism. Moreover, the degradation of the proline hydroxylated and acetylated HIF-1α may be considered as rate-limited in tumors [29]. The fact that severe hypoxia is a driving force for HIF-1α stabilization and metabolic reprogramming has been considered a central process in cancer development for a long time [30]. This opinion is in accordance with earlier studies from Papandreou and co-authors, demonstrating that HIF-1α stimulates glycolysis and represses mitochondrial function and oxygen consumption, by inducing the pyruvate dehydrogenase kinase 1 to inhibit pyruvate dehydrogenase from using pyruvate required for the mitochondrial TCA cycle [31].

The sophisticated HIF-1α/pVHL system is well-balanced and regulated by different molecular events. First, different missense mutations in the α- and β-domain of the *VHL* gene cause an altered function of pVHL [32]. In addition, the accumulation of HIF-1α through loss-of-function mutations or promoter hypermethylation in the *VHL* gene leads to a dramatic reprogramming of OSCC, due to the prevention of the ubiquitin E3 ligase complex linked degradation [20,33].

Despite the importance of HIF-1α stabilization caused by the functional loss of VHL, polymorphisms or mutations in HIF-1α can increase its activity [34]. Furthermore, deletions or polymorphisms in the tuberosis complex (TSC) 1 and TSC2 could result in abnormal accumulation of HIF-1α [35]. Moreover, changes in the HIF-1a coding sequence, and a change in the ODD domain that converts Pro-582 to Ser, have no effect on VHL-binding after hydroxylation at proline 564 [36]. Some cytokines have also been shown to be involved in the stabilization of HIF-1α. Noteworthy, interleukin (IL)-6 and IL8 as well as tumor necrosis factor (TNF)-α contribute to the progression of OSCC [37]. In particular, IL-8 is the most important factor for malignant transformation in OSCC [38]. The phenomenon of a HIF-1α mediated stabilization under normoxia upregulating IL-1β has also been published [39]. Consequently, HIF-1α acts as a central transcription factor linking inflammatory and oncogenic pathways [39].

The HIF expression and regulation, in which various feedback mechanisms are involved, is very complex and thus far is not completely understood. This metabolic modulation could be inhibited by different substances. For example, Wei and coworkers demonstrated a general downregulation of HIF-1α and phosphatidylinositol-4-5-bisphosphonate 3 kinase (PI3K) by salvianolic acid B in an experimental oral cancer model [40]. Thus, salvianolic acid B may have anti-carcinogenesis potential in OSCC [40].

Despite the fact that the shift from oxidative phosphorylation (OXPHOS) to glycolysis, also called ‘glycolytic’ shift, is a crucial hallmark of oral cancer [41], tumors exhibit a great heterogeneity and cancer cells could reverse from glycolysis to OXPHOS, thereby maintaining both reprogrammed and oxidative metabolism in the same tumor [41]. This general phenomenon, called cancer cell metabolic plasticity, is a consequence of a highly heterogeneous TME influenced by diverse factors, such as oxygen concentration, nutrient and energy supply and the extent of vascularization and blood flow. This is controlled by a crosstalk of oncogenic signaling, transcription factors and growth factors, as well as reactive oxygen species.

Among the immune checkpoints, B7-H3 affects the proliferation, migration and invasion of OSCC [42]. Furthermore, B7-H3 promotes the WARBURG expression effect, as measured by an increased glucose uptake such as lactate production due to an upregulation of HIF-1α, which is mediated via the PI3K/Ant/mTOR pathway [42].

In sum, the activation of HIF-1α can be considered as a complex process, in which different tumor suppressor genes as well as various oncogenes are involved [43].

## 3. HIF-1α Stabilization under Normoxia, Its Role in Glutamine Metabolism, Low Extracellular pH and Epithelial Mesenchymal Transition

It is generally accepted that HIF-1α is a central transcription factor in various cancer types including OSCC [44]. For many years, our research team focused on the analysis of both hypoxic, but also normoxic stabilization of HIF-1α in this disease [26,45,46]. HIF-1α is upregulated in OSCC when oxygen consumption outstrips its supply [8,9,47]. Thus, HIF-1α is the first driving force for cancer development, organizing the energy supply in OSCC under normoxic as well as hypoxic conditions [26,46,48]. So far, the stabilization of HIF-1α is poorly understood. However, a stabilization of HIF-1α under normoxia in the presence of glutamine and growth factors was found in OSCC cell lines [26,48]. In contrast, the inhibition of glutaminolysis reduced the accumulation of HIF-1α under normoxic conditions. Ammonia/ammonium, a toxic waste product of glutaminolysis, induced the normoxic stabilization of HIF-1α [48,49], which is also mediated by a reduced acetylation at the residue Lys 532 [25,26]. The mechanisms, by which hypoxic stress and/or the HIF system shift the glutamine metabolism from oxidation to reductive carboxylation, was identified. HIF-1 activation significantly reduced the activity of the α-ketoglutarate dehydrogenase as a key mitochondrial enzyme complex [50]. In addition, a HIF-dependent expression of the *SLC1A1* and *SLC1A3* genes, encoding glutamate transporters, and of the *GRIA2* and *GRIA3* genes, encoding glutamate receptors, was reported [51,52]. Thus, HIF-1α is sufficient to activate important signal transduction pathways that promote cancer progression [52].

Otto Warburg postulated an enhanced conversion of glucose to pyruvate, and consequently to lactate, even in the presence of abundant oxygen [20,21,53]. This HIF-1α-dependent adaptation includes a “selfish” reprogramming process mediated by an overexpression of glucose transporters [21]. Indeed, the overexpression of the glucose transporter (GLUT)-1 negatively interferes with the prognosis and overall survival (OS) of OSCC patients [54]. In addition to GLUT-1, GLUT-4 also enables glucose transport to the tumor cell, thereby replacing the oxidative phosphorylation with glycolysis as an energy source in OSCC [55]. GLUT-4 was identified in human gingiva [56] and determined a malignant phenotype, suggesting a role of this protein in oral carcinogenesis [55]. Malignant cells require increased glucose transport to organize their energy support and an overexpression of GLUT family members are characteristic features [57]. Moreover, other molecules involved in glycolysis mediated by HIF-1**α** have been described. The lactate dehydrogenase (LDH)-5 catalyzes the conversion of lactate to pyruvate [58]. Furthermore, LDH-5 is also expressed in saliva, so that salivary LDH-5 content reflects the biological behavior and aggressiveness of OSCC, and might therefore serve as an additive prognostic factor in OSCC [58].

A major target protein upregulated by HIF-1α stabilization is the carbonic anhydrase (CA) IX catalyzing the reaction between protons and CO_2_ to H_2_CO_3_. In the metabolic adaptation by the HIF-mediated pathway, CA IX as a central target protein of the HIF system plays an important role [45]. An elevated expression of CA IX was associated with a poor prognosis of OSCC patients [59] and with a 5.13-fold increased risk of tumor-related death (*p* = 0.017) [60]. This was confirmed by Li and coworkers, suggesting its use as a prognostic biomarker for this disease [61,62,63]. Thus, CA IX was established as an additive prognostic marker [45,64]. In addition, a role of CA IX as a pH sensor was established, and the first therapeutic options by the selective inhibition of carbonic anhydrases, leading to a reversion of the multidrug resistance in tumor cells, were reported [65].

Next to CA IX, a prognostic role of hexokinase 2 (HK2)—the speed limiting key enzyme of glycolysis—was found with a more than two-fold increased risk of tumor-related death due to a HK2 overexpression in OSCC (Berg T and co-authors, unpublished results). Consequently, the enhanced glucose uptake enables tumor cell growth, which was connected with an accumulation of lactate, thereby adapting the intracellular pH (pH_i_) levels. An overview regarding the complex system of intercellular pH stabilization has recently been summarized by Kappler and coworkers [46].

Epithelial mesenchymal transition (EMT) represents a key step in the development of metastases in OSCC. Loss of VHL expression significantly correlated with the EMT process of OSCC, with β-catenin and HIF-1α as downstream mediators [66]. Indeed, HIF-1α is known to regulate a large group of genes/proteins involved in the regulation of the cellular metabolism, pH and EMT [17]. A pivotal role of HIF-1α, to organize pH_i_ stability followed by a lower extracellular pH (pH_e_), was demonstrated in OSCC [46]. To stabilize the pH value, a HIF-1α-dependent upregulation of CA IX is necessary [45]. A disturbed extracellular matrix orchestrated by HIF-1 was found as an important factor in EMT of hypoxic cells [67]. Thus, the extracellular acidification might be another driving force to adapt the E-cadherin/vimentin balance. A loss of E-cadherin in combination with an overexpression of vimentin are early steps in the EMT cascade [68]. The acidification of the extracellular matrix and the maintenance of an optimal pH for sustaining cancer growth has been well investigated. In addition to the CA IX expression, a number of other additive molecules are upregulated. These include the monocarboxylate transporters (MCTs), Na^+^/H^+^ exchangers (NHEs), natrium bicarbonate cotransporters (NBCs), vacuolar ATPases (V-ATPases), and some anion exchangers (AEs) [69]. The expression of these proton exchangers can be used as potential prognostic biomarkers and therapeutic targets in OSCC. The transport of waste products as a consequence of glucose and glutamine consumption is regulated by MCTs. Elevated levels of MCT1 and MCT4 are associated with a poor prognosis of OSCC patients [70]. Their function in stabilizing the pH_i_, as well as the acidification of the TME, supports the EMT machinery. The overexpression of proton exchangers in order to acidify the extracellular matrix by lowering the pH_e_ enables the maintenance of optimal pH_i_ for sustaining cancer growth and aggressiveness in OSCC (Figure 2). Repression of protein exchangers is a promising tool to enhance anti-tumor therapy [69]. Both proton transporters and exchangers act synergistically and represent key features for tumor cells to survive in a very hostile microenvironment.

In conclusion, HIF is able to trigger various pathways, which may lead to a more aggressive tumor type characterized by metastases and EMT promotion [71]. The lowering of pH_e_ might represent a crucial role in the induction of the EMT cascade, while EMT also causes the activation of transcription factors, reorganization/expression of cytoskeletal proteins, upregulation of extracellular-matrix-proteins and others [71,72].

## 4. Translational Aspects of HIF-1α in Oral Cancer Development

To determine the role of HIF-1α in OSCC, the expression of HIF-1α was analyzed in large cohorts of OSCC lesions by various groups. Ribeiro and coauthors investigated 93 OSCC samples for HIF-1α expression, demonstrating that metastatic lymph nodes and intra-tumoral regions of corresponding primary tumors (91.7%, *p* = 0.001) expressed HIF-1α at a high frequency [73]. Qian and coworkers described a prognostic role of both HIF-1α and HIF-2α isoforms for OSCC development, which was associated with an increased risk of mortality in this kind of cancer [74]. The HIF-1α expression pattern could be used to discriminate healthy mucosa, precursor lesions and OSCC. Furthermore, epithelial dysplastic lesions with an increased HIF-1α expression had a greater risk of malignant transformation, suggesting that the expression of HIF-1α is an early event in oral carcinogenesis [75]. Moreover, HIF-1α could be employed to characterize aggressive subtypes in early OSCC stages. The prognostic significance of HIF-1α overexpression in OSCC was evaluated in a meta-analysis, suggesting that HIF-1α is the independent prognostic marker in OSCC patients [18]. We also found a correlation between an overexpression of HIF-1α and poorer patients’ survival [15], which was in contrast to a report by Fillies and coauthors [76]. This discrepancy might be explained by the half-life of only a few minutes for HIF-1α [77]. To sum up, the expression of HIF-1α might serve as an additive prognostic tool in the diagnosis and therapy selection of this disease [78].

## 5. MicroRNAs Involved in the Metabolic Alterations of OSCC

Non-coding RNAs (ncRNAs) are characterized as a group of RNAs without any capacity to encode proteins [79,80] and consist of microRNAs (miRs) and long ncRNA [80]. Short 18–22 base pair long endogenous miRNAs play a key role in controlling gene expression by regulating several physiological, but also pathological processes, including carcinogenesis [81]. In addition, miRNAs have been identified as participating in the regulation of metabolic remodulating [82]. In this context, a number of hypoxia and HIF-dependent miRNAs with oncogenic and/or suppressive functions in cancer development were summarized [80,83,84,85]. These condensed data suggest that hypoxia is a master regulator of miRNA genes.

To date, more than 1600 different miRNAs have been identified, from which some have already been associated with OSCC in general and might serve as a potential biomarker [86,87]. Furthermore, miR-21, miR-455-5p, miR-155-5p, miR-372, miR-373, miR-29b, miR-1246, miR-196a, miR-181 and miR-210 are upregulated by hypoxia in OSCC, while miR-204, miR-101, miR-32, miR-20a, miR-16, miR-17, and miR-125b are downregulated under hypoxic conditions [45,81]. Within these hypoxia-regulated miRNAs, miR-21 is one of the most significantly upregulated miRNAs. In addition, miR-21 upregulation markedly enhanced snail and vimentin expression and significantly decreased E-cadherin levels in OSCC [88], suggesting that miR-21 is not only HIF-associated, but is also a relevant initiator for EMT, promoting prometastatic properties in this disease (Figure 3). MiR-210-3p, a member of the miR-210-family, is also involved in HIF-mediated adaptations [83]. Its overexpression contributed to an accelerated OSCC growth and revealed that HIF-1α specifically interacted with the promoter of miR-210-3p, thereby enhancing its expression [89]. Thus, miR-210 has been considered as a master hypoxamiR, regulating the expression of multiple target genes in order to fine-tune the adaptive response of cells to hypoxia [90]. In addition, hypoxia and/or normoxic HIF-1α modulates the activity of key proteins that control posttranscriptional events in the maturation and activity of miRNAs [85]. Thus, hypoxia-associated miRNAs are integrated into a complex hierarchical regulatory network induced by oxygen demand and hypoxic stress. The complex crosstalk between HIF-dependent and HIF-independent regulatory pathways and a role of miRNAs in hypoxic cellular adaptations has been described [85]. As a consequence, miR-210 is involved in metabolic reprogramming, DNA repair and cell cycle arrest, vascular biology and angiogenesis [83] whereas miR-21 is a key regulator of apoptosis.

## 6. Conclusions

The metabolic adaptions in OSCC are versatile. This article reflects the major metabolic changes in this disease. One hallmark of OSCC development is the normoxic as well as hypoxic stabilization of HIF-1α as a general regulator to overcome metabolic stress and energetic disorganization. This involves more than 1000 target genes of HIF-1α, which organize the energy metabolism. In addition, HIF-1α induced the stabilization of the pH_i_ to secure tumor cell survival and DNA replication. Consequently, due to the sophisticated adaptation system mediated by CA IX and transmembrane transport molecules, the extracellular pH (pH_e_) is decreased and initiates the EMT cascade. Some metabolism-controlled miRNAs are involved in this complex adaptation process, in particular members of the miR-200 family. Moreover, the extracellular acidification has been shown to be involved in the alteration of anti-tumoral in immune responses. However, the immunological adaptations to the acidic TME are beyond the scope of this review and will be discussed in detail in an accompanying publication.

## Figures and Tables

**Figure 1 ijms-21-06083-f001:**
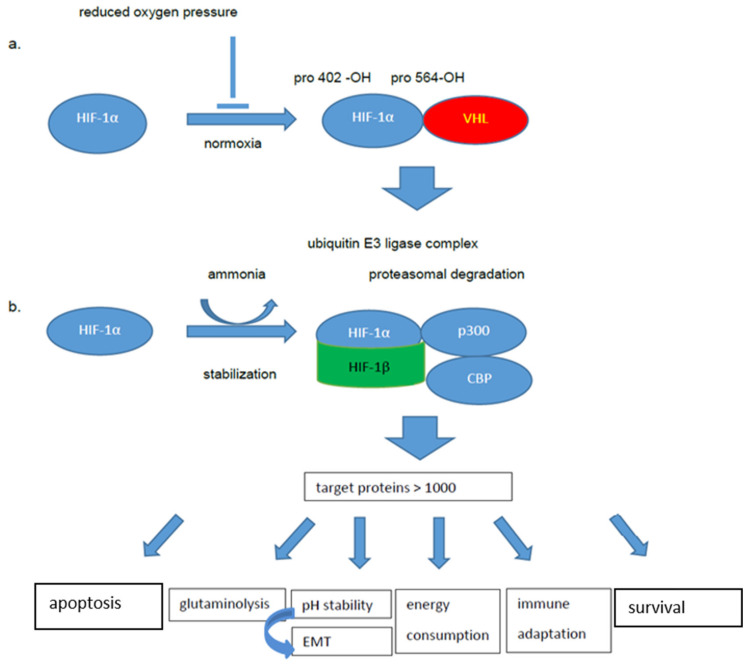
Role of HIF-1α in oral carcinogenesis. (**a**). Classical normoxic condition. Hydroxylation at proline residues 402 and 564, dimerization with the VHL protein and proteasomal degradation due to a ubiquitin E3 ligase complex. (**b**). Hypoxia of oral cancer and “tumor normoxia”, ammonia and/or low pH levels prevent the proline hydroxylation and can cause the stabilization of HIF-1α, its transport to the nucleus, dimerization with HIF-1β, which in combination with the p300 protein and CBP complex results in the regulation of more than 1000 target genes.

**Figure 2 ijms-21-06083-f002:**
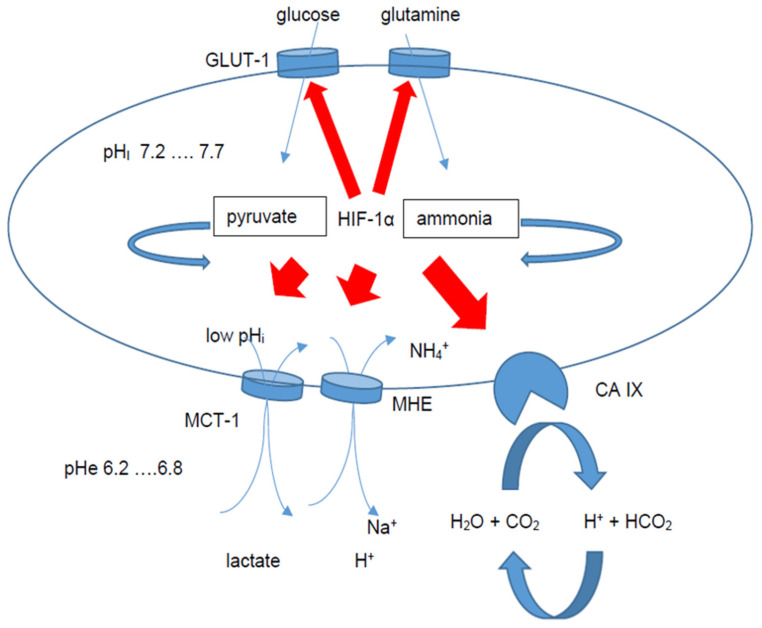
Metabolic adaptation to an increased energy consumption for enhanced tumor cell proliferation. HIF-1α activates the glucose uptake and glutamine utilization. Toxic waste products, protons and ammonia induce an intracellular acidification accompanied by a decrease in pHi. As a consequence, HIF-1α also induces an upregulation of NHEs and MCTs as well as CA IX. CA IX catalyzes the reaction of protons and hydrogencarbonate to H2O + CO2. The pHe is more acidic and acts as a driving force in the early EMT process. Red arrows show significant HIF-1α-dependent processes and upregulated proteins. Blue arrows show catalyzing steps by enzymes as well as transport across the tumor cell membrane.

**Figure 3 ijms-21-06083-f003:**
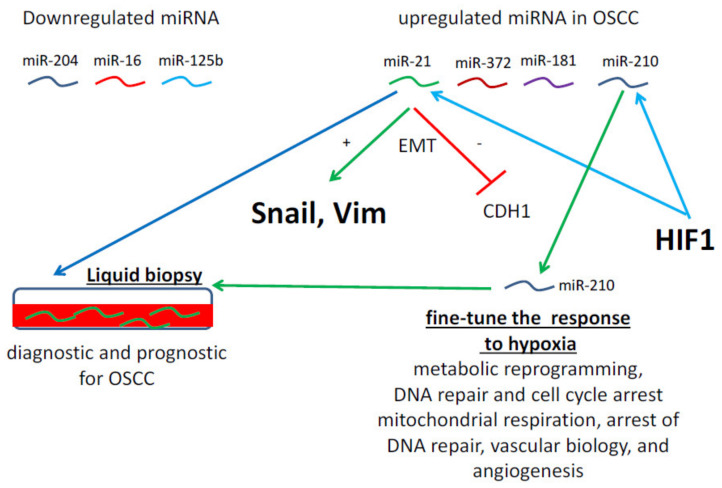
Deregulated miRNAs in OSCC. Some representative deregulated miRNAs in OSCC are given, which are hypoxia/HIF1-regulated or involved in EMT. The use of miRNAs present in liquid biopsies might be used as diagnostic or prognostic markers in OSCC.

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
