# Peer review of "Current Understanding of the HIF-1-Dependent Metabolism in Oral Squamous Cell Carcinoma"

_ijms, 2020, doi:10.3390/ijms21176083_

Round 1
Reviewer 1 Report
The manuscript submitted to the International Journal of Molecular Sciences by A.W. Eckert et al. entitled “Current understanding of the HIF1 dependent metabolism in oral squamous cell carcinoma.” Is an overview of the features of the HIF1 pathway and downstream targets implicated in tumorigenesis, cancer progression in the context of metabolism in oral squamous cell carcinoma (OSCC). It is good that this review is focused on pH and metabolism regulation; however, I notice few gaps.
- The authors mention HIF-1a stabilization in normoxic conditions, but did not mention any mutations present in OSCC which cause constitutive HIF1a protein expression. Very frequently this is what initiates the tumor.
- It is also well established that cytokines stabilize HIF1a in normoxia, and IL-1, IL-6, IL-8 and TNFa are upregulated in OSCC, hence it should be mentioned as well.
- The authors mention tumor microenvironment but hardly comment on that in the context of hypoxia/HIF axis and metabolism.
- The abstract says that therapy response will be addressed, but nothing was mentioned.
I have few other suggestions and comments:
- Figure 1 – please make sure the figure is more readable and more organized; I am confused about the arrow (“U-turn arrow”) just above glutaminolysis; more biological processes regulated by the HIF targets should be listed (i.e. apoptosis, survival…); also possibly add the arrow from pH stability causing EMT
- Line 85 says that HIF-1 regulates more than 40 target genes (probably early publication on HIF), but it is known that the number is much higher >1,000 and even the authors mentioned it.
- “These include the monocarboxylate transporters (MCTs), Na+/H+ exchangers (NHEs), natrium bicarbonate cotransporters (NBCs), and vacuolar ATPases (V-ATPases) [60] (Lorenzo-Pouso et al. 2019)” this sentence has been repeated in the same paragraph..?
- Figure 2 – please improve the figure description. CA IX catalyzes the reaction of protons and hydrogencarbonate to H2O + CO2 and vice versa, so the arrow should be pointing both ways. Explain the meaning of red and blue arrows.
- Please double check the grammar and acronyms throughout the manuscript; number of abbreviations was introduced and not really mentioned later on.
Author Response
Reviewer 1
Critism
- The authors mention HIF-1a stabilization in normoxic conditions, but did not mention any mutations present in OSCC which cause constitutive HIF1a protein expression. Very frequently this is what initiates the tumor.
Answer
We included a chapter dealing with mutations in VHL-gene with respect to normoxic HIF-stabilization
Critism
- It is also well established that cytokines stabilize HIF1a in normoxia, and IL-1, IL-6, IL-8 and TNFa are upregulated in OSCC, hence it should be mentioned as well.
Answer
The role of some cytokines – including IL-1, IL-6, IL-8 and TNFa upregulated in OSCC was discussed and references included
Critism
- The authors mention tumor microenvironment but hardly comment on that in the context of hypoxia/HIF axis and metabolism.
Answer
We complemented this chapter by including some other aspects of EMT-initiation (transcription factors…)
Critism
- The abstract says that therapy response will be addressed, but nothing was mentioned.
Answer
The last sentence in abstract was changed from “therapeutic response” to “therapeutical stratification” with any reflections in the mean text
other suggestions and comments:
- Figure 1 – please make sure the figure is more readable and more organized; I am confused about the arrow (“U-turn arrow”) just above glutaminolysis; more biological processes regulated by the HIF targets should be listed (i.e. apoptosis, survival…); also possibly add the arrow from pH stability causing EMT
Answer
The figure 1 is now more readable; the arrow (“U-turn arrow”) above glutaminolysis is explained below and also more biological processes regulated by the HIF targets are included: apoptosis and survival
- Line 85 says that HIF-1 regulates more than 40 target genes (probably early publication on HIF), but it is known that the number is much higher >1,000 and even the authors mentioned it.
Answer
We corrected all lines to more than 1000 target genes induced by HIF
- “These include the monocarboxylate transporters (MCTs), Na+/H+ exchangers (NHEs), natrium bicarbonate cotransporters (NBCs), and vacuolar ATPases (V-ATPases) [60] (Lorenzo-Pouso et al. 2019)” this sentence has been repeated in the same paragraph..?
Answer
The repeated sentence has been omitted.
- Figure 2 – please improve the figure description. CA IX catalyzes the reaction of protons and hydrogencarbonate to H2O + CO2 and vice versa, so the arrow should be pointing both ways. Explain the meaning of red and blue arrows.
Answer
The figure description is improved; red and blue arrows explained – the arrow is pointed in both ways concerning CA IX
- Please double check the grammar and acronyms throughout the manuscript; number of abbreviations was introduced and not really mentioned later on.
Answer
The manuscript was checked to grammar and acronyms; all abbreviations are mentioned in the text
Reviewer 2 Report
The submitted manuscript is a review article focused on regulation of HIF-1α-mediated metabolic reprogramming in oral squamous cell carcinoma (OSCC). The topic is of great interest of scientific audience, since HIF-1 is a master regulator of gene expression responsible for metabolic reprogramming of cancer cells under hypoxia conditions observed on many cancers including OSCC. However, there are some concerns and recommendations, which are as follows.
Major concerns:
The title of the manuscript is “Current understanding of the HIF1 dependent 2 metabolism in oral squamous cell carcinoma”, which implicates that all HIF-1-dependent metabolic pathways involved in carcinogenesis should be considered. However, the participation of HIF-1 in expression of genes encoding glycolytic enzymes and those involved in involved in other metabolic pathways such as TCA cycle and lipid metabolism has not been discussed. More comprehensive discussion of literature data is needed. The Conclusions section stated that “The article reflects the major metabolic changes in this disease”; however, not all important HIF-1-mediated changes in enzyme activities involved in OSCC were discussed. See PMID: 31737114, PMID: 26577856, PMID: 29789538. PMID: 32252351 and others.
Since the title starts from the words “Current understanding” the authors should discuss current viewpoint on cancer cell metabolism, metabolic heterogeneity of cancer cells, i.e. interplay between oxidative and reprogrammed metabolism.
3. The manuscript style is poor since the same ideas and sentences are repeated throughout the text and sentences have short and sketchy character.
Minor concerns:
1. There are two sections with the same title “The HIF-1α system as a general regulator in OSCC”. Moreover, the first of them, which encompasses lines 64-94 discusses regulation of HIF-1 activity and its target genes, so the title should be changed. Additionally, the next section (lines 95-118) with the same title discusses HIF-1stabilization under normoxia conditions and its role in glutamine metabolism, therefore the title should be corrected.
2. Figure 1 has to be moved from Introduction to next section.
3. Lines 124-125: Findings of Quin et al.: which isoforms of HIF-1?
4. “Metabolic adaptation” section, lines 137-165. The title of the section implicates csncer cell adaptation through various metabolic changes; however, here only carbonic anhydrase IX catalytic activity-mediated response to HIF-1 stabilization in OSCC cells was discussed.
5. Figure 2 contains GLUT-1-mediated glucose uptake by cancer cells. Indeed, unlike normal cells, where GLUT-4 is a major glucose transporter, in cancer cells GLUT-1 is a primary glucose transporter; and this should be discussed in the text in more details.
6. References in the text are given in both numbers and author names; this should be corrected.
7. Grammar and style: title and elsewhere: HIF-1; line 24: “which functions as” instead of “ as functions as”; line 38 “hallmarks” instead of “hallmark”; line 47: “HIF-1α’ or “HIFs” instead of “HIF”; line 145: there should be “ (CA) IX catalyzing reaction between protons and”; line 188: “acidification” instead of “ acidify”.
Author Response
Reviewer 2
Critism
Major concerns:
The title of the manuscript is “Current understanding of the HIF1 dependent 2 metabolism in oral squamous cell carcinoma”, which implicates that all HIF-1-dependent metabolic pathways involved in carcinogenesis should be considered. However, the participation of HIF-1 in expression of genes encoding glycolytic enzymes and those involved in involved in other metabolic pathways such as TCA cycle and lipid metabolism has not been discussed. More comprehensive discussion of literature data is needed. The Conclusions section stated that “The article reflects the major metabolic changes in this disease”; however, not all important HIF-1-mediated changes in enzyme activities involved in OSCC were discussed. See PMID: 31737114, PMID: 26577856, PMID: 29789538. PMID: 32252351 and others.
Since the title starts from the words “Current understanding” the authors should discuss current viewpoint on cancer cell metabolism, metabolic heterogeneity of cancer cells, i.e. interplay between oxidative and reprogrammed metabolism.
Answer
We included some more important HIF-1-mediated changes in enzyme activities involved in OSCC. The suggested PMID´s : 31737114, PMID: 26577856, PMID: 29789538. PMID: 32252351 were discussed and included in the manuscript.
Critism
- The manuscript style is poor since the same ideas and sentences are repeated throughout the text and sentences have short and sketchy character.
Answer
The manuscript style was improved – repeated sentences and headings has been omitted to eliminate the short and sketchy character
Minor concerns:
- There are two sections with the same title “The HIF-1α system as a general regulator in OSCC”. Moreover, the first of them, which encompasses lines 64-94 discusses regulation of HIF-1 activity and its target genes, so the title should be changed. Additionally, the next section (lines 95-118) with the same title discusses HIF-1stabilization under normoxia conditions and its role in glutamine metabolism, therefore the title should be corrected.
Answer
The same title of two sections was eliminated, moreover, the title of the next section reflects normoxic conditions as well as the pivotal role in glutamine metabolism. The Title was changed in “ HIF-1 α – stabilization under normoxia, its role in glutamine metabolism, low extracellular pH and epithelial mesenchymal transition“
- Figure 1 has to be moved from Introduction to next section.
Answer
Figure 1 has been moved to the next section in the manuscript.
- Lines 124-125: Findings of Quin et al.: which isoforms of HIF-1?
Answer
We explained the isoforms HIF-1α and HIF-2α in this section
- “Metabolic adaptation” section, lines 137-165. The title of the section implicates csncer cell adaptation through various metabolic changes; however, here only carbonic anhydrase IX catalytic activity-mediated response to HIF-1 stabilization in OSCC cells was discussed.
Answer
We discussed the role of additional glucose transporters (GLUT-3/-4) with respect to energy metabolism, pH-stability and early effects in EMT
- Figure 2 contains GLUT-1-mediated glucose uptake by cancer cells. Indeed, unlike normal cells, where GLUT-4 is a major glucose transporter, in cancer cells GLUT-1 is a primary glucose transporter; and this should be discussed in the text in more details.
Answer
We discussed the role of other glucose transports (GLUT-3/-4) and their role in oral carinogenesis and included references
- References in the text are given in both numbers and author names; this should be corrected.
Answer
All author names in the text has been eliminated
- Grammar and style: title and elsewhere: HIF-1; line 24: “which functions as” instead of “as functions as”; line 38 “hallmarks” instead of “hallmark”; line 47: “HIF-1α’ or “HIFs” instead of “HIF”; line 145: there should be “ (CA) IX catalyzing reaction between protons and”; line 188: “acidification” instead of “ acidify”.
Answer
Grammar and style has been brushed.
Altogether, the reviewer comments enhanced the quality of the manuscript. We hope it is now acceptable for publication in the International Journal of Molecular Sciences.
Kind regards,
Alexander Eckert
Corresponding author
Round 2
Reviewer 2 Report
The manuscript has been significantly improved. However, there are some typos, which have to be corrected. Most of them are indicated in the text in blue and red colors (see attached file). Please, carefully, ckeck the manuscript grammar.

Author Response
First review:
Critism
Major concerns:
The title of the manuscript is “Current understanding of the HIF1 dependent 2 metabolism in oral squamous cell carcinoma”, which implicates that all HIF-1-dependent metabolic pathways involved in carcinogenesis should be considered. However, the participation of HIF-1 in expression of genes encoding glycolytic enzymes and those involved in involved in other metabolic pathways such as TCA cycle and lipid metabolism has not been discussed. More comprehensive discussion of literature data is needed. The Conclusions section stated that “The article reflects the major metabolic changes in this disease”; however, not all important HIF-1-mediated changes in enzyme activities involved in OSCC were discussed. See PMID: 31737114, PMID: 26577856, PMID: 29789538. PMID: 32252351 and others.
Since the title starts from the words “Current understanding” the authors should discuss current viewpoint on cancer cell metabolism, metabolic heterogeneity of cancer cells, i.e. interplay between oxidative and reprogrammed metabolism.
Answer
We included some more important HIF-1-mediated changes in enzyme activities involved in OSCC. The suggested PMID´s : 31737114, PMID: 26577856, PMID: 29789538. PMID: 32252351 were discussed and included in the manuscript.
Critism
- The manuscript style is poor since the same ideas and sentences are repeated throughout the text and sentences have short and sketchy character.
Answer
The manuscript style was improved – repeated sentences and headings has been omitted to eliminate the short and sketchy character
Minor concerns:
- There are two sections with the same title “The HIF-1α system as a general regulator in OSCC”. Moreover, the first of them, which encompasses lines 64-94 discusses regulation of HIF-1 activity and its target genes, so the title should be changed. Additionally, the next section (lines 95-118) with the same title discusses HIF-1stabilization under normoxia conditions and its role in glutamine metabolism, therefore the title should be corrected.
Answer
The same title of two sections was eliminated, moreover, the title of the next section reflects normoxic conditions as well as the pivotal role in glutamine metabolism. The Title was changed in “ HIF-1 α – stabilization under normoxia, its role in glutamine metabolism, low extracellular pH and epithelial mesenchymal transition“
- Figure 1 has to be moved from Introduction to next section.
Answer
Figure 1 has been moved to the next section in the manuscript.
- Lines 124-125: Findings of Quin et al.: which isoforms of HIF-1?
Answer
We explained the isoforms HIF-1α and HIF-2α in this section
- “Metabolic adaptation” section, lines 137-165. The title of the section implicates csncer cell adaptation through various metabolic changes; however, here only carbonic anhydrase IX catalytic activity-mediated response to HIF-1 stabilization in OSCC cells was discussed.
Answer
We discussed the role of additional glucose transporters (GLUT-3/-4) with respect to energy metabolism, pH-stability and early effects in EMT
- Figure 2 contains GLUT-1-mediated glucose uptake by cancer cells. Indeed, unlike normal cells, where GLUT-4 is a major glucose transporter, in cancer cells GLUT-1 is a primary glucose transporter; and this should be discussed in the text in more details.
Answer
We discussed the role of other glucose transports (GLUT-3/-4) and their role in oral carinogenesis and included references
- References in the text are given in both numbers and author names; this should be corrected.
Answer
All author names in the text has been eliminated
- Grammar and style: title and elsewhere: HIF-1; line 24: “which functions as” instead of “as functions as”; line 38 “hallmarks” instead of “hallmark”; line 47: “HIF-1α’ or “HIFs” instead of “HIF”; line 145: there should be “ (CA) IX catalyzing reaction between protons and”; line 188: “acidification” instead of “ acidify”.
Answer
Grammar and style has been brushed.
second Revision
We rechecked the language and grammar another one by all authors. The typos are eliminated. We hope, our manuscript is now acceptable for publication.